# Electronic Alert Signal for Early Detection of Tissue Injuries in Patients: An Innovative Pressure Sensor Mattress

**DOI:** 10.3390/diagnostics13010145

**Published:** 2023-01-01

**Authors:** Jinpitcha Mamom, Bunyong Rungroungdouyboon, Hanvedes Daovisan, Chawakorn Sri-Ngernyuang

**Affiliations:** 1Center of Excellence in Creative Engineering Design and Development, Faculty of Engineering, Thammasat University, Pathum Thani 12121, Thailand; 2Department of Adult Nursing and the Aged, Faculty of Nursing, Thammasat University, Pathum Thani 12121, Thailand; 3Human Security and Equity Centre of Excellence, Social Research Institute, Chulalongkorn University, Bangkok 10330, Thailand; 4Institute of Field Robotics, King Mongkut’s University of Technology Thonburi, Bangkok 10140, Thailand

**Keywords:** biomarkers, early detection, electronic alert signal, innovative mattress, pressure sensors, patients with tissue injuries

## Abstract

Monitoring the early stage of developing tissue injuries requires intact skin for surface detection of cell damage. However, electronic alert signal for early detection is limited due to the lack of accurate pressure sensors for lightly pigmented skin injuries in patients. We developed an innovative pressure sensor mattress that produces an electronic alert signal for the early detection of tissue injuries. The electronic alert signal is developed using a web and mobile application for pressure sensor mattress reporting. The mattress is based on body distributions with reference points, temperature, and a humidity sensor to detect lightly pigmented skin injuries. Early detection of the pressure sensor is linked to an electronic alert signal at 32 mm Hg, a temperature of 37 °C, a relative humidity of 33.5%, a response time of 10 s, a loading time of 30 g, a density area of 1 mA, and a resistance of 7.05 MPa (54 N) at 0.87 m^3^/min. The development of the innovative pressure sensor mattress using an electronic alert signal is in line with its enhanced pressure detection, temperature, and humidity sensors.

## 1. Introduction

Early detection of tissue injuries can be facilitated by a crucial pressure sensor for lightly pigmented skin damage in patients. The early stage of cell damage in patients comes with the risk of developing pressure injuries. Moreover, treating patients with tissue injuries are complicated. The complexity is due to cell damage, the breakdown of skin, deep tissue injury, and cell death [1,2,3]. The global treatment rate of tissue injuries in patients is approximately 14.8%, and it is 6–18.5% in the acute clinical setting. Emergency care accounts for 6.31% [4]. Treatment costs for tissue injuries in patients can range from USD 894.69 to USD 98,730.24 per year [5].

Despite the development of diagnostic devices, it remains difficult to detect the occurrence of early cell damage in patients [6,7]. For early detection of lightly pigmented skin injuries in patients [8,9], pressure sensors provide an accurate reading. The characteristics of lightly pigmented skin in patients are defined as ‘wet-bulb and dry-bulb cells’, and an accurate pressure sensor is required. Early tissue injuries are detected based on the first visual sign of skin damage, defined as the ‘heralding sign’, to report the cutaneous blanche response [10]. For early detection devices, an electronic signal alert enables precise detection from the pressure sensor on lightly pigmented skin injuries in patients.

An electronic alert signal from a pressure sensor on lightly pigmented skin damage plays an important role in detecting native skin colour, cell diagnosis, and magnetic resonance [11,12]. Previous studies have indicated that early detection of darkly pigmented skin damage relies on accurate biosensor applications [13,14,15]. An accurate method for early detection requires a real-time alert, mechanical loading, effective monitoring, and applicable diagnostic devices. A highly innovative mattress that can accurately detect early cell damage in patients is appropriate [16]. However, temperature and humidity sensors are integral to the development of an accurate pressure sensor mattress.

Early detection of lightly pigmented skin injuries in patients requires a highly force-sensitive sensor, a fast response, an accurate diagnosis, and reliable sensing devices [3,6,10,14,17]. Previous studies have investigated the biophysical response to invisible cell damage during the tumour development of tissue injuries [18]. Some studies have suggested that temperature and humidity sensors are paramount in detecting tissue injuries [19,20]. There is recent research on cell organs, 3D printed diagnoses, body pressure distributions, and blood pressure tests [21,22,23,24]. Lightly pigmented skin damage is a key focus in terms of early detection. However, current temperature and humidity sensors used for intact skin surface are inadequate.

In the subepidermal layer, various studies have found moisture in patients with a dark skin tone [7,8,14,25]. Recent studies have obtained measurements involving temperature and the photonic, electrical, ultrasound, biofilm, and dermal fluids [26,27,28,29,30]. To date, there is limited research on detecting temperature and humidity in patients with lightly pigmented skin injuries [6,21,31,32]. The purpose of the current study is to develop an innovative pressure sensor mattress using an electronic signal alert for early detection of tissue injuries in patients. We aim to examine perfusion and blood circulation (wet and sweat) using temperature (°C) and humidity (%) sensors.

## 2. Literature Review

### 2.1. Tissue Injury Stages

Tissue injuries are the cause of cell damage in blood vessels; they also result in sprains, strains, chemokines, contusions, and tendon issues [33,34]. Maver et al. [35,36] declared that the causes of tissue injuries include an acute trauma, a chronic wound, an infection, and genetic disorders. Tissue injuries are typically found in regions where early cell damage occurs adjacent to bony prominences, as shown in Figure 1.

*Lightly pigmented skin injuries (stage I)*. Such an injury is defined as the first visible change in the skin and is known as the ‘heralding sign’.*Darkly pigmented skin injuries (stage II).* This type of injury is defined as partial-thickness skin loss with an exposed dermis.*Blanchable erythema injuries (stage III).* This type of injury is defined as full-thickness skin loss, such as adipose, granulation tissue, and epibole.*Pressure injury with oedema (stage IV).* This injury is defined as full-thickness skin and tissue loss with exposure of directly palpable fascia, muscle, tendon, ligament, cartilage, and bone in the ulcer.

### 2.2. Tissue Injury Detection

Early detection of lightly pigmented skin injuries in patients is effective for the prevention of cell damage. Previous studies have defined four stages of tissue injuries: non-blanchable erythema of intact skin (stage I), partial-thickness skin loss with exposed dermis (stage II), full-thickness skin loss (stage III), and full-thickness tissue loss (stage IV) [37]. Previous studies have classified tissue injury as ‘a localized purple or maroon area of discoloured intact skin due to damage of underlying soft tissue from pressure and/or shear’ [38]. According to Kolluru et al. [39], patients with lightly pigmented skin injuries exhibit painful, firm, mushy, boggy, and warmer/cooler areas. In the present study, we detect patients with lightly pigmented skin damage and theorise that this is an effective region to implement an electronic alert signal from an accurate pressure sensor mattress.

### 2.3. Temperature Detection

The temperature of patients with tissue injuries plays a crucial role in the early detection of lightly pigmented skin damage. According to Kokate et al.’s [40] definition, ambient temperature is detected on the skin surface of patients’ body organs, such as the heart, liver, brain, and blood. The temperature detection is based on electronic vibrations, elevation of rectal temperature, antioxidative enzyme activity, and the formation of malondialdehyde [41]. The temperature ranges from 1 to 100 mm Hg, withstands a strain of 26%, and detects cycles of 40 °C, and the tissue injury has a silky substrate of 1.5 μm thick epoxy. As temperature increases by 1 °C, which relates to 1 mm Hg, there is a force-sensitive detection of 10%. The temperature ranges from 35 to 40 °C, with a change of ±0.1 °C from 37 °C to 39 °C and of ±0.2 °C when below 37 °C and above 39 °C [42].

Previous studies have suggested a temperature-increase (<1°C) resonance-based passive detection (>2 moisture), an accuracy of ±2%, and a pressure loading time of 10–15 s for the early detection of temperature [43]. The temperature of tissue injuries in patients ranges from 30 to 50 °C, has a sensitivity of <0.1 °C, and may increase by 94% for early detection [44]. However, it is unclear whether a patient’s temperature, even at a high level, may overcome the limitation of detection. It is unknown how to detect lightly pigmented skin injuries [45]. To reinforce this point, a previous study on temperature detection in patients found a range of 35–40 °C, with a pressure sensor of 32 mm Hg and an accuracy of ±1% [46].

### 2.4. Humidity Detection

Humidity in patients with tissue injuries is defined as subepidermal moisture-induced cell damage from water in the epidermal and dermal tissues [47]. One type of humidity is erythema at the buttocks, ischia, trochanters, sacrum, and coccyx with dark tones [48]. Humidity in patients with tissue injuries is associated with hydrated edges of the hand, abdomen, thighs, legs, and lower back [49]. Schwart et al. [50] suggested that humidity can be detected in the skin’s moisture, such as sweat, urine, and saline. There have been efforts to detect patients’ damaged layers deeper in the skin through the sinking-in of the probe and the bulging of adjacent skin from mechanical injuries to the plantar tissues.

Gefen et al. [51] found that humidity involves visual skin assessment, moisture, and skin-water vapour. To ensure humidity detection, the pressure sensor is assessed in the dry-bulb and wet-bulb thermometer moistures [52]. The dry-bulb and wet-bulb tissues involve the thermodynamic activity of air and water vapour. Bates-Jensen et al. [16] suggested that humidity is detected via the non-blanchable erythema (early redness) of skin tone. It is important to note that moisture, wetness, lipids, and porcine are responsible for humidity detection in patients with lightly pigmented skin injuries.

## 3. Materials and Methods

### 3.1. Detection Procedures

The electronic signal alert procedures used to detect lightly pigmented skin injuries are combined with pressure detection (mm Hg), temperature (T), and humidity (H) [53]. The detection procedures of tissue injuries provide an accurate response to perfusion, change of position, and moisture. The parameter is connected to oxygen, tissue-perfusion, temperature, and humidity sensors. The lightly pigmented skin has not suffered from ischemic damage as a result of non-contact, which is not reported. The diagram depicting the detection procedures is displayed in Figure 2.

### 3.2. Pressure Sensors

The pressure sensors for tissue injuries in patients were developed using a force-sensitive resistor to test static measurement and repeatability. The sensor array was based on the supine and side-lying positions and had a pressure loading of 32 mm Hg [54]. The subjects were in a side-lying position, adjusted from 25 to 40 mm Hg in a supine region. The temperature ranged from 35 to 40 °C. Humidity ranged from 10 to 100%. They varied with the detection of the electronic alert signal. The force-sensitive sensor was converted to an alert signal on the dashboard monitor. The electronic signal took the form of a transmitted microcontroller in the (yes) pin and the (no) pin. The pressure sensor is illustrated in Figure 3.

### 3.3. Developing an Innovative Pressure Sensor Mattress

Our development of an innovative pressure sensor mattress for early detection of tissue injuries incorporated temperature and humidity data. Figure 4 depicts the development of the pressure sensor mattress. The detection array is set at 4.37 mm, with two sensors spanning across the platforms. The array consists of three functional layers: pressure detection, temperature, and humidity sensors. Infrared probes are placed in the supine and side-lying positions. The force-sensitive sensor is set at 1200 kΩ, with a loading time range of 10–30 s and a normal force range from 10 to 30 g.

### 3.4. The Functional Sensing System

The functional system includes three arrays. First, the pressure sensors detect data and time. Second, the temperature sensors detect skin, time, and average temperature. Third, the humidity sensors account for skin wetness and dryness. Figure 5 details the functional mechanism of the mattress. The array includes four top and four bottom force sensors. All sensors receive the detection data, which are then displayed on the microcontroller. The sensor database is subsequently sent to the dashboard monitor of the pressure sensor mattress. Figure 6 presents the pad sensor in the detection mechanism. The datasheet consists of two force-sensitive sensor boxes as follows:The sensor is mounted on a hard surface;The contact pad is smaller than the sensitive area;The contact pad is mounted in the central array;Permanent loads are not applied to the sensor to avoid drift;The sensor is bent in the active area.

### 3.5. Control System

The key control systems are as follows: (i) temperature and humidity sensors, (ii) pressure detection, (iii) microcontroller (input and output), and (iv) dashboard monitor (web and mobile app). The pressure sensor mattress rays correspond to a microcontroller (+ or –out). The signal alert reporting system is connected to the dashboard monitor via Wi-Fi. The mechanism consists of automatic and user controls. The temperature and humidity sensors are connected to a digitalised monitor for reporting.

Figure 7 depicts the control system of the electronic signal mechanism. The system consists of four temperature and humidity sensors and eight pressure sensors that are connected to the microcontroller monitor. The signalling data are sent to the electronic alert system. The force-sensitive system is displayed in Figure 8. The sensor is mounted on the FSR sensor, which measures the interface pressure for the side-lying position. The data-based signal is sent to the pressure sensor mattress via the dashboard monitor.

### 3.6. Dashboard Monitor

The dashboard monitor is displayed in Figure 9. The sensor includes seven contacts of epoxy on the dashboard monitor. The outputs are as follows: (i) automatic system, (ii) display of pressure, (iii) operational system, (iv) temperature and humidity data, (v) conditional notification, (vi) real-time detection, and (vii) keypads. The seven key elements of the microcontroller include the following:➀Real-time displays and reporting with command keypads ‘on’ the left and right after a specification time;➁A display of the patient’s weight and surveillance alerts;➂A flip command to control the screen;➃A display of the real-time temperature and view in period;➄A display of the magnitude of the force-sensor hazard when the pressure is over 32 mm Hg, with a red alert warning;➅A display of the temperature and humidity values;➆Keypad command (autos and manual control).

## 4. Results

### 4.1. Patients with Tissue Injuries

The database test for the pressure sensor mattress included 30 patients suffering from tissue injuries at the Thammasat University Hospital, Thammasat University Rangsit Campus, Pathum Thani, Thailand. All patients provided informed consent to participate in the study. We adhered to the Declaration of Helsinki, the Belmon Report, the CIOMS Guidelines, and international practice (ICH-GCP) (COA 122/2564 Project No.121/2564). The thirty patients were divided into two groups (*n* = 15 experimental group and *n* = 15 control group). The tests were conducted every two hours in a four-week trial from 1–29 December 2021. Table 1 and Table 2 display the characteristics of the experimental and control groups.

### 4.2. Platform of Mattress

The platform included pressure detection regions, temperature, and humidity sensors. The test ranged from 1 to 15 MPa for the online converter of weights, which was used to measure tissue injuries in patients. The temperature ranged from 35 to 40 °C. The humidity (skin wetting and abnormal skin effect) ranged from 10 to 50% for lightly pigmented skin injuries. The loading and unloading times were set at 32 mm Hg to examine the stability of the accurate pressure detection. The outputs were recorded before and after the sensor responses.

Figure 10 depicts the platform of the mattress. The platform was tested using patient information (gender, age, ID code, and disease) on the dashboard monitor. Figure 11 presents the structure of the dashboard monitor. The screen displays the pressure detection, temperature, and humidity sensors, which are linked to the electronic alert signal on the dashboard monitor. The monitor can be set as an automatic system. It includes a timer, the ability to turn left and right, and/or a handling screen.

### 4.3. Pressure Sensor Test

Static calibrations of the pressure sensor were used for the test; this included eight temperature and humidity sensors and four pressure detections. The force-sensitive resistor was consistently set at 1.52 cm in width and 60.96 cm in length, with a distance of 5 cm. The force resistance was set at 1200 kΩ, and the range was from 10 to 100 kΩ. The loading time was set at 30 g, which changed from 1 to 5 g. The nominal resistance accounted for 12%, and the fast response sensor was between 10 and 30 s.

Figure 12 illustrates the body sensors of the experimental group. It shows the supine area of the body’s angle at 45° from the body, a midstance at 15° from the femur, a sagittal angle at 30° dorsal, and a frontal angle at 45° medial. The left-lying and right-lying positions are set at 90° from the body sensor (sagittal angle at 15° ventral, frontal angle at 30° medial, and sagittal angle at 45° dorsal). The force-sensitive detection of tissue injuries in patients is presented in Table 3.

Force-sensitive resistors are used to test the pressure detection and loading time (g). The force-sensitive resistor is based on a weight set at 1.96 N of the loading time on the voltage divider circuit as follows:(1)$$V_{0}=V_{cc}\left(\frac{R}{R+FSR}\right)$$
where $V_{0}$ is the output of voltage, $V_{cc}$ is the input of voltage, and *R* is the pulldown resistance. The resistance labelled ‘*R*’ is divided into an equation with a set of 1200 kΩ resistors, which can reach as low as 100 kΩ when force pressure is applied.

The experimental test using weights is shown in Figure 13. Regarding kΩ, the fastest response time of sensors 2 and 3 ranges from 2.59 to 5.83% of probability. There is a 1.44% probability that sensors 1, 4, and 5 would be misclassified as sensor 6. The FSR drifts are plotted on the loading pressure (5–10 g) and are fitted to the sensor for 10 s. There is a difference of almost five times compared to pressure loading (30 g), allowing the system to monitor respiratory rates up to 1000 kΩ per 10 s.

### 4.4. Temperture and Humdity Sensor Tests

The temperature and humidity tests relied on internal factors (gender, age, and disease) and the external factor of mm Hg. The humidity ranged from 20 to 50% and had an accuracy of ±5% relative humidity. The temperature ranged from 35 to 40 °C, and the accuracy was ±2 °C. The test was repeated at 0.5–2% of the relative humidity. The detection had a fast response (<10 s), a resolution of <1%, a tolerance of ±10%, and a loading time of 30 s.

Table 4 includes a comparison of the temperature and humidity sensors. The normal temperature remains stable at 40 °C and is recorded every 10 s. The temperature is repeated for the RTD240.5 ± 0.0 reference sensor. The response time of RTD is over 10 times that of 264.0 ± 0.0 and has the average 1/$e_{rise}$ (30 s). The values show that the H steps of 10% RH are then adjusted to 10–70% RH, keeping T constant at 7.5 ± 0.5 °C. We also compared the variability of the reference to RTD260.1 ± 0.2 of T (35 °C) at RH55.0 ± 0.4, defined as the time to change from 10% to 70% of the final values.

Figure 14 illustrates the temperature test. It was recorded using a high-resistance probe sensor placed 10 cm into the rectum, with an accuracy of +0.2 °C and 3-point calibration. The test showed that sensors 1 and 2 of T1, T2, and T3 were placed in areas of the body (lying position). The sensor achieves detection at 35.5 °C and 27 mm thickness. The T4 and T5 of sensors 3 and 4 are accurately detected within ±10%. They also withstand a temperature gradient reaching 37.5 °C. The results (Figure 15) highlight that the humidity remains constant at 36.5% and that the moduli range from 37.5 to 42.5% RH. The validity of the results was monitored using a hygrometer (±5% accuracy) of 64.4 ± 0.2 with 42.5% increases every 30 s, respectively.

### 4.5. Force Sensor Test

The standard loading time (SLT) is defined as skin reddening in the same subjects and regions of a patient’s position. The subjects refer to the areas of pressure, and there is a set of 4 kPa over 30 min for the SLT. The regions refer to the pre-damaged skin, including sheer forces, excessive muscular activities, and injuries on the lumbar spine. The SLT is a set of$\;10^{-4}\;{\mathrm{kPa}}^{-4}$, with a resolution of$\;10^{5}$ in the static test of 0.1 kPa. The SLT is calculated for the pressure *p* as follows:SLT (min) = 2400/*p* (kPa)(2)

The SLT is applied as follows: (i) the skin is covered by thin layers of fat and muscle; (ii) the skin has a highly convex curvature; (iii) the skin is not trained by cyclic loading; (iv) the patient is at risk according to the Norton scale; and (v) the skin is pre-damaged. When the SLT is suited for the sensor, the layers are pressed on the vapour of *P* data (*p* = 1.0 to 95.2, 96.0, or 96.7 kPa). The SLT is created (*p* = 1.0 to 50.0 kPa) for skin damage, with a set of 43 kPa at 32 mm Hg for the equal components.

The maximum of the force sensor is set to ±1.95 MPa at <1 N. The temperature ranges from 35 to 40 °C. The humidity ranges from 7.5 to 40% on patients’ lightly pigmented skin injuries. The support factor (SUF) is provided as $P_{s}$. Whether $P_{s}$ is suitable for the loading data is defined as follows:(3)$$\mathrm{SUF}=/P_{m}$$
where the $P_{m}$ is the maximum average of pressure loading. The SLT and SUF are compared with the subject regions in the supine and the side-lying positions. The parameter has a drop in discharge piping of 10%, a normal relief of 10%, and an accumulation of 20%. The fracturing-based fluid is a set of 8.17$\;m^{3}$ at 6.13 MPa and 0.92 $m^{3}$/min. The cross-link fracturing of the fluid is 22.37 $m^{3}$. The flow rate is equivalent to 7.05 MPa at 0.87 $m^{3}$/min. The safety factor (*S_F_*) is defined as follows:(4)$$S_{F}=\frac{R_{p0.2\min}}{\sigma_{\mathrm{redB}}}\geq \;1.0$$
where $S_{F}$ = safety factor against bolt yield pressure load;

$\sigma_{\mathrm{redB}}$ = $\sqrt{\sigma_{2}^{2}+3\;(k_{\tau }\tau }{)}^{2}$ = von Mises equivalent pressure;

$\sigma_{z}$ = $\frac{1}{A_{S}}$ $(F_{\mathrm{Mzu}}+$ $F_{\mathrm{SAmax}})$;

τ = $\frac{16M_{G}}{{\pi d}_{S}^{3}}$;

$k_{\tau }$ = pressure reduction factor = 0.5.

The parameter is the maximum rate of error in the detection on lightly pigmented skin injuries. The four subjects are the following: (i) the small and light type (150 cm/60 kg); (ii) the high and light type (170 cm/80 kg); (iii) the small and heavy type (155 cm/99 kg); and (iv) the high and heavy type (175 cm/100 kg). The maximum rate is 330 min for the high and light types and 920 min for the small and light types, with a pressure average (${\mathrm{AV}}_{m}$) of 750 min, a range parameter (${\mathrm{RP}}_{m}$) of 1.25, and causes of error at 5%. For the small and light types, ${\mathrm{RP}}_{m}$ = 1.58 and the ${\mathrm{AV}}_{s}$ is 500 min. Table 5 presents a comparison of the averages, ranges, and loading times.

The force-sensitive sensor was pressed in the supine position to test the stability of the loading time. The test consisted of pressure loading (10–60 g) in 10–30 s. The resistance ranges from 200–1200 kΩ, and there is a value of 1.95 MPa (15 N). The sensor is based on loading and unloading, and the pressure sensor is on the mattress screen. Since the loading time is appropriate, the SUF is 50% higher in a supine position. Figure 16 displays the tests of the kΩ and SLT.

The SLT and the SUF have results of 10% for the pressure sensor, 36% for the tall and light types, and 72% for the small and heavy types. For the force-sensitive subjects,$\;{\mathrm{RP}}_{m}$ = 2.07 for the small and light types and ${\mathrm{RP}}_{m}$ = 1.48 for the high and light types. The SUT increases by 30%, which appears to recover at ≈0.1 s of loading and 0.1 s of unloading time. A sensor set of 10 g (15 N) was applied to the loading and unloading cycles (0.5 Hz). Figure 17 shows the resistance and weight loading.

### 4.6. Static Test

The static calibration is based on the force-sensitive sensor outputs. The different weights are recorded to ascertain the actual drift of voltage, loading time, and signal detection. Static calibration is formulated as follows:(5)$$Drift\left(t\right)\frac{V_{FSR}\left(t\right)-V_{FSR}\left(0\right)}{V_{FSR}\left(0\right)}$$
where the *Drift(t)* is a normalised voltage; $V_{FSR}$*(t)* is the loading time; *t*, and $V_{FSR}$(0) is the pressure detection. Each sensor which is based on weight loading ranges from 0.059 to 18.8 N at 117 voltage in the centre-side position. The resistor $R_{G}$ is 1200 kΩ as the voltage $V_{\mathrm{FSR}}$ across the sensor is 1 V. The loading time is a set of 10 s, ranging from 10 to 30 g, with a repeated test of 10 times for signal detection. The loading time is combined with a 1.3 × $10^{-}$^4^ V/Pa of 0.9 kPa in 1 h for the force sensor. Table 6 sets the criteria for the force-sensitive resistor.

Figure 18 presents the static sensor dataset of 15 kPa on the mattress. The temperature ranges from 30 to 40 °C. The relative humidity ranges from 30 to 40%. The loading time is 12 in d, and the absolute drift is 32 mm Hg. The pressure loading is a set of 18 N, 100 kΩ, ${\mathrm{RP}}_{m}$ = 1.25, and loading response of 10–30 s. The pressure loading is 10 g at 0.954 in a supine position for lightly pigmented skin injuries.

### 4.7. Repeated Temperature, Humidity, and Pressure Sensor Test

The pressure sensor is the core pressure loading of 32 mm Hg, temperature at 30–40 °C, and RH at 10–50%. According to the repeated tests, the loading time is 30 g, 25 N, 1000 kΩ, ${\mathrm{RP}}_{m}$ = 1.75, 1.90 kPa, with fluid of 32.24 $m^{3}/\min$. The force signal is detected at a temperature of 36 °C, RH at 33.5%, and 32 mmHg. The detection was repeated (+3%) with hysteresis (10%), a response time of 30 s, an active area of 12.7 mm, and a density area of 1mA. Figure 19 depicts the repeatability of temperature and RH. Figure 20 includes a test of the repeatability of the pressure, temperature, and humidity sensors.

## 5. Discussion

### 5.1. Discussion with Rsults

We aim to develop innovative pressure detection, temperature, and humidity sensors for an electronic signal mattress. The innovative pressure sensor mattress demonstrates early detection at 32 mm Hg, a pressure loading of 30 g, a loading time of 10 s, a temperature of 36 °C, a RH of 33.5%, hysteresis of 10%, 25 N, and 1000 kΩ. Our data confirm the results of prior studies that the detection of tissue injuries is significant for patients with lightly pigmented skin damage (13,25,31). Based on our findings and the results of prior research, early detection (with electronic signal alerts) of tissue injuries in patients is effective in the clinical care setting [2,6,9,14,24,25,36,40,42].

The temperature and humidity sensors show the most active early detection, flexible substrate deforms, and resistance [55,56]. The force-sensitive resistors are converted into electronic signals for early detection of lightly pigmented skin injuries in patients [2,32]. The loading time of 10 s with pressure loading at 30 g and fluid at 32.24 $m^{3}/\min$ are appropriate components of a pressure sensor mattress. This finding is consistent with a study conducted by Lee et al. 2019 [57], who found the pressure loading signal could be detected at 10–30 s at 34.3 mm Hg. Previous studies have discovered that high damping of pressure loading needs to be linked to the accuracy of early detection in lightly pigmented skin injuries [58,59].

The electronic signal mattress shows the greatest early detection at 32 mm Hg, with 25 N, 1000 kΩ, ${\mathrm{RP}}_{m}$ = 1.75, and 1.90 kPa. This finding is supported by the previous studies by Yu et al. [60] and Malmsjö et al. [61], which indicated that a pressure loading at 32 mm Hg may increase early detection by 10% in patients. A pressure sensor mattress is shaped to produce an electronic signal alert for early detection of tissue injuries in patients in a diagnostic setting [3,20,47]. The advantage of the temperature and humidity sensors is that they are gentle on a directly detected signal alert system with a loading time at 10 s of both high sensitivity and high specificity (66% and 72%, respectively). Even with limited data (*n* = 15 experimental groups and 15 control groups), the pressure sensors show promise in early detection for the clinical utility of the mattress.

Proper measurement from eight temperature and humidity sensors and four pressure sensors is useful for early detection in the first stage of skin damage [13,57,62,63]. The temperature (35–40 °C), humidity (37.5–42.5%), and loading time of 10 s are the most significant electronic alert signals of the pressure sensors. The measurement shows that temperature increases by 0.5 °C, humidity increases by 50%, and an electronic alert signal is detectable at 20 s. This finding is consistent with previous work, which has shown that temperature sensors are effective for early detection in the first stage of tissue injuries [14,16,20,24,51]. To date, temperature sensors have incorporated detection responses when applied to lightly pigmented skin damage [64].

The innovative pressure sensor mattress is the most effective method in the early detection of electronic alert signals based on the loading time (on) and unloading time (off) procedure [6,8,11,14,24,35,65,66]. The electronic signal alert of the web and mobile application is a proactive prognosis system in clinical care settings. We found early detection with a loading time of 10 s at 32 mm Hg and ${\mathrm{RP}}_{m}$ = 1.75, an accurate time within ±10%, a temperature gradient of up to 37 °C, and a relative humidity of 33.5%. The results show that an accurate pressure sensor can be used for detection in temperature and humidity sensors.

### 5.2. Practical Diagnostics

This study has some practical diagnostic devices. First, the accurate pressure sensors have been validated with a set of 32 mm Hg; this may increase early detection in 94% of patients. Even in a high-performing diagnostic setting, an electronic signal is usually detected only once or twice. Second, the signal detection yields noteworthy accuracies (loading time of 10 s at 30 g), indicating that such a rapid routine could be used for reliability in the developmental stage. Third, the pressure sensor intricately interacts with ambient data, indicating that the electronic signal potentially causes alert responses in real time. There are also the factors of stable temperature and humidity sensors, which are mostly automatic, and the flow of liquid through early signal detection. Apart from usability, the temperature and humidity sensors are capable of detecting a temperature of 37 °C and RH of 33.5% on tissue injuries. Correct detection, early signal operation, and pressure sensor capability in the context of the mattress solution have developed a fully integrated diagnostic system with electronic signal alert.

### 5.3. Innovation Device Contributions

This research is novel because the pressure sensor mattress, along with the temperature and humidity sensors, can be used to detect tissue injuries. This innovative diagnostic device implies that tissue injuries in patients can be detected freely while performing daily activities. This innovative mattress (as shown in Figure 7) with real-time electronic signal detection (see Figure 12) is now available to optimise the operation and usable devices. This is a diagnostic mattress with sufficient sensitivity (as indicated in Figure 20), and the developer requires a medical launch of an innovative mattress. New medical devices, diagnostic procedures, detectable sensors, prognostic sensitivities, and electronic signals are being produced. This process creates an additional burden for hospital administrators, clinical engineers, and medical staff who are responsible for the acquisition of new technology.

As a result, there is a pressing need for medical engineers to assume more responsibilities in these two areas. First, new innovative mattress technology must include an evaluation of safety, efficacy, and cost-effectiveness, as well as a consideration of the social, legal, and ethical effects of these diagnostic devices. Second, for nursing care institutions to remain cost-effective and competitive, a new medical mattress can be selected based on the knowledge gathered about its performance, value, and availability. The processes of innovation, development, and diffusion of public and private medical technology play important roles in advancing diagnostic mattress technology.

## 6. Conclusions

To conclude, we presented an innovative pressure sensor mattress that produces an electronic alert signal for the early detection of tissue injuries in patients, yielding an accuracy that exceeds the previous report. This study reveals an electronic signal alert of 32 mm Hg, along with results for temperature (37 °C), relative humidity (33.5%), pressure loading (30 g), and density area (1mA). Effectiveness studies testing medical devices under conditions resembling real-world practice can be improved with the use of correct diagnostic medical devices. Within this development, an electronic pressure sensor mattress with electronic signal alert and pressure detection, temperature, and humidity sensors can have end-user application for detecting tissue injuries in patients. This mattress is implemented for early detection with existing sensor devices, which can be used in clinical care settings.

### Limitations and Future Direction

This study has some limitations. First, the abovementioned diagnostic device is the prototype of an innovative mattress using electronic alert signals and pressure detection, temperature, and humidity sensors. The experimental group only includes 30 patients with tissue injuries. Second, the pressure sensor mattress is still in the developmental stage. Through the development and application of innovative pressure sensor mattresses, future studies are needed to gain widespread clinical acceptance of these diagnostic devices. The current study provides the platform upon which to further improve these medical devices to enable patient-specific diagnostics for those with tissue injuries. Once available, it will be used as a diagnostic device and an example of innovative pressure sensor mattresses in real-world clinical care settings.

## Figures and Tables

**Figure 1 diagnostics-13-00145-f001:**
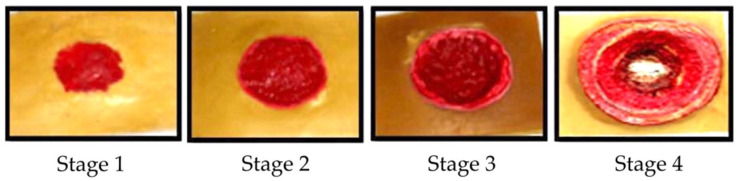
Stages of tissue injuries.

**Figure 2 diagnostics-13-00145-f002:**
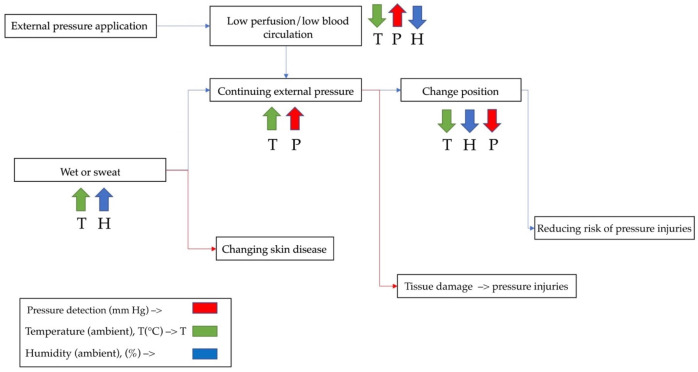
Detection procedures.

**Figure 3 diagnostics-13-00145-f003:**
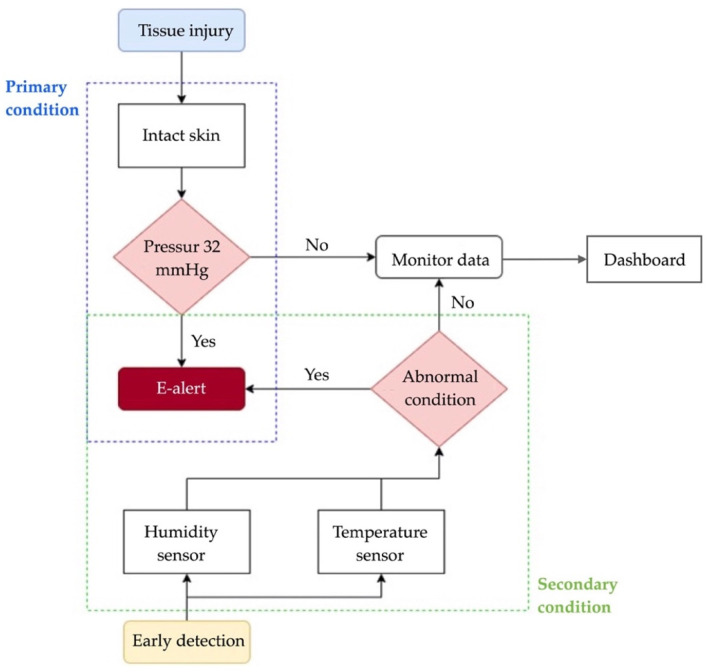
Diagram of the pressure sensors.

**Figure 4 diagnostics-13-00145-f004:**
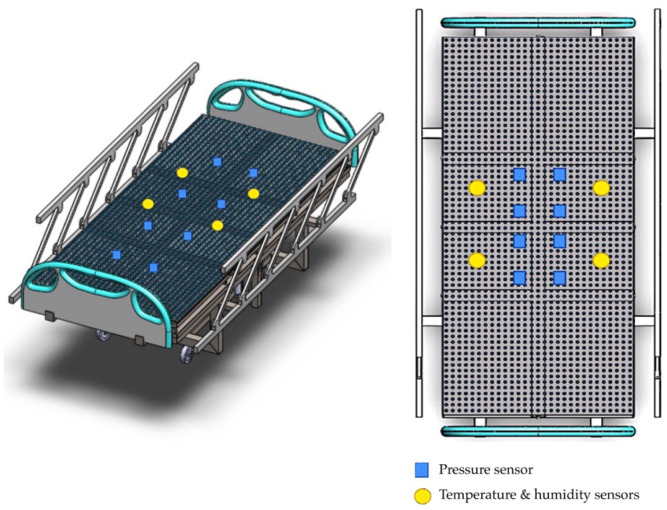
Development of the mattress.

**Figure 5 diagnostics-13-00145-f005:**
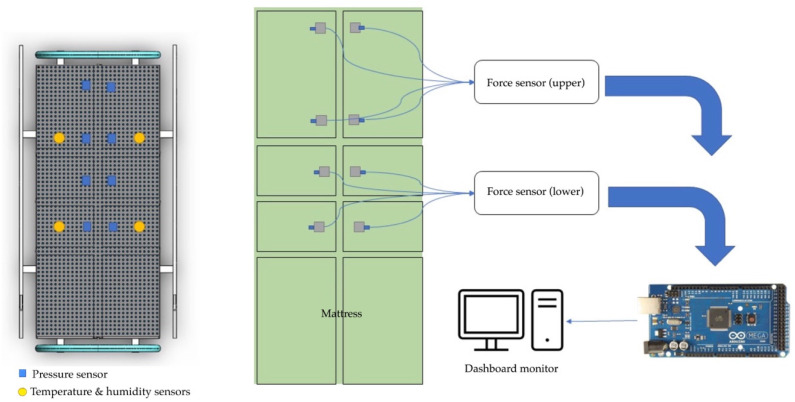
Functional mechanism.

**Figure 6 diagnostics-13-00145-f006:**
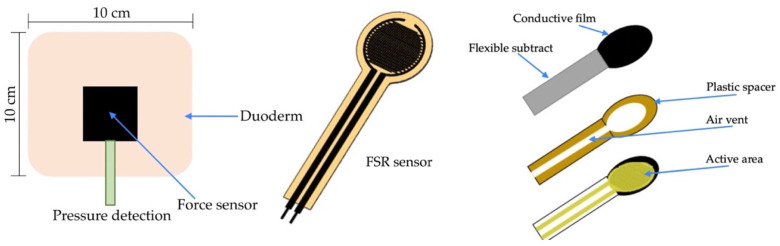
Pad sensor.

**Figure 7 diagnostics-13-00145-f007:**
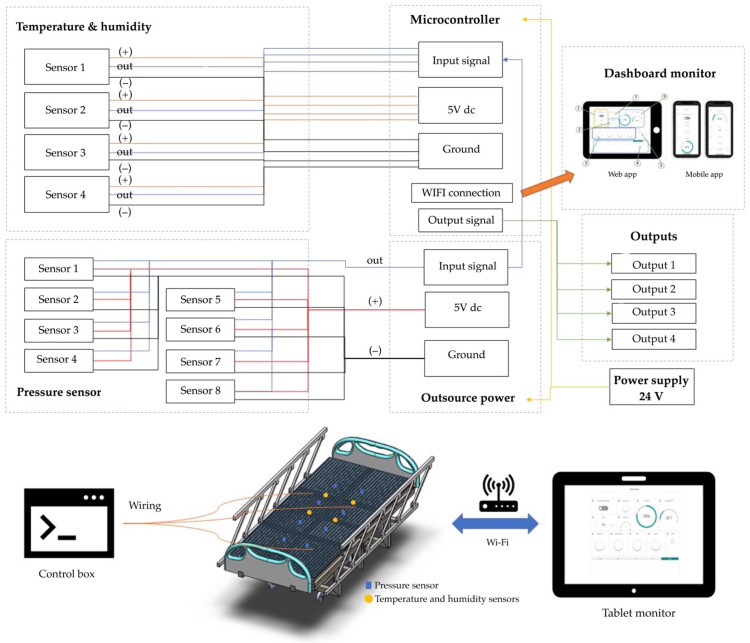
Control system of the electronic signal mechanism.

**Figure 8 diagnostics-13-00145-f008:**
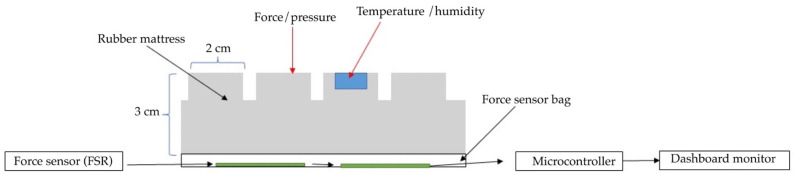
Force-sensitive mechanism.

**Figure 9 diagnostics-13-00145-f009:**
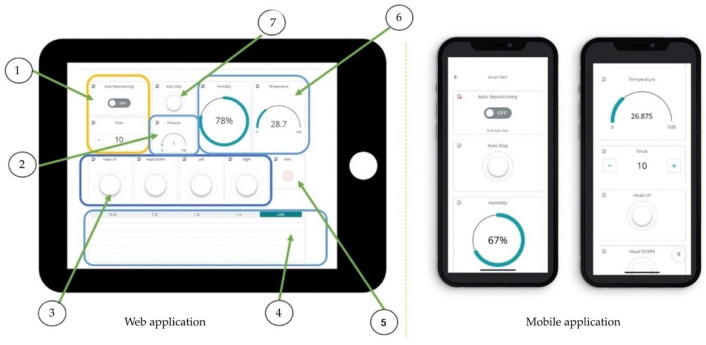
Web and mobile application.

**Figure 10 diagnostics-13-00145-f010:**
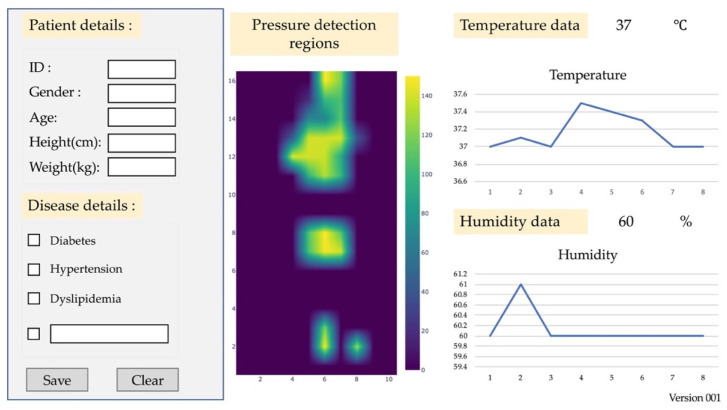
Mattress platform.

**Figure 11 diagnostics-13-00145-f011:**
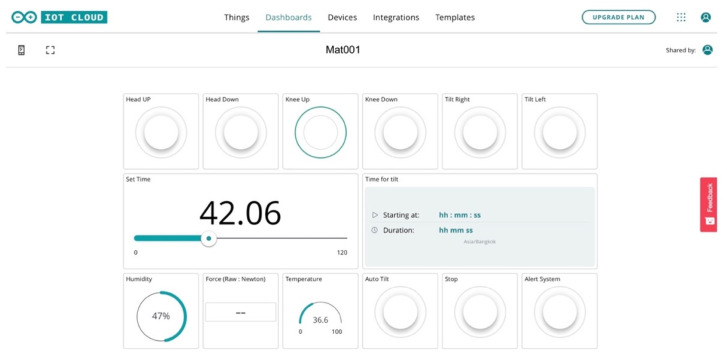
Dashboard monitor structure.

**Figure 12 diagnostics-13-00145-f012:**
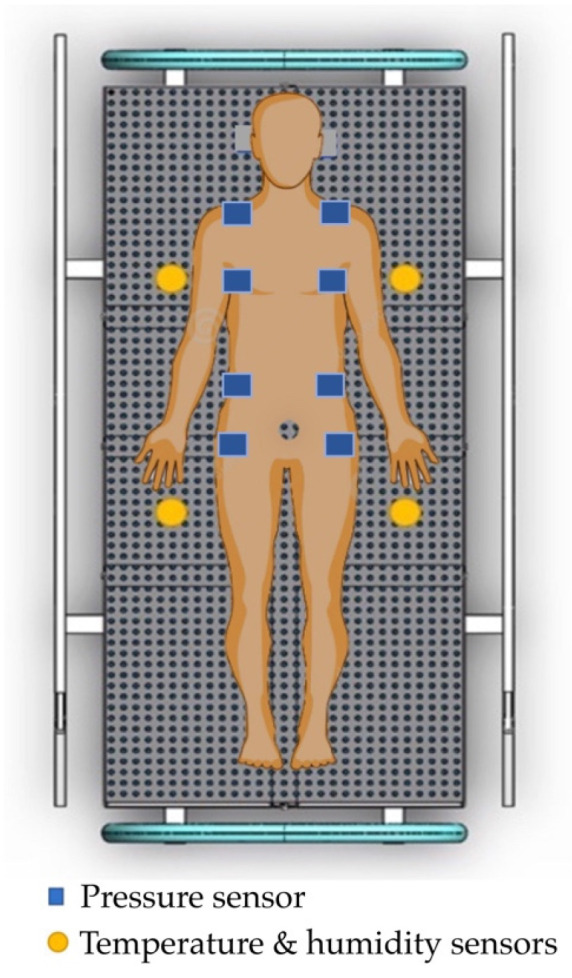
The body sensors.

**Figure 13 diagnostics-13-00145-f013:**
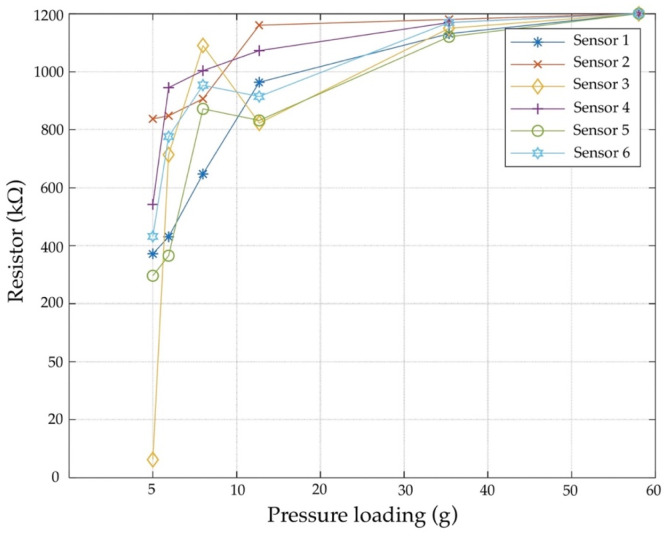
Pressure sensor response test.

**Figure 14 diagnostics-13-00145-f014:**
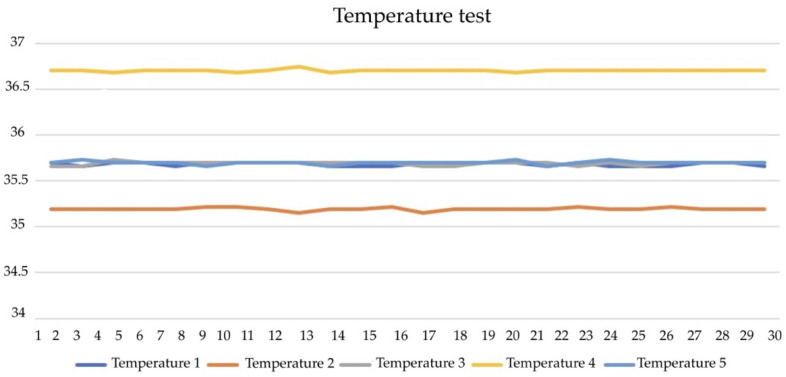
Temperature test.

**Figure 15 diagnostics-13-00145-f015:**
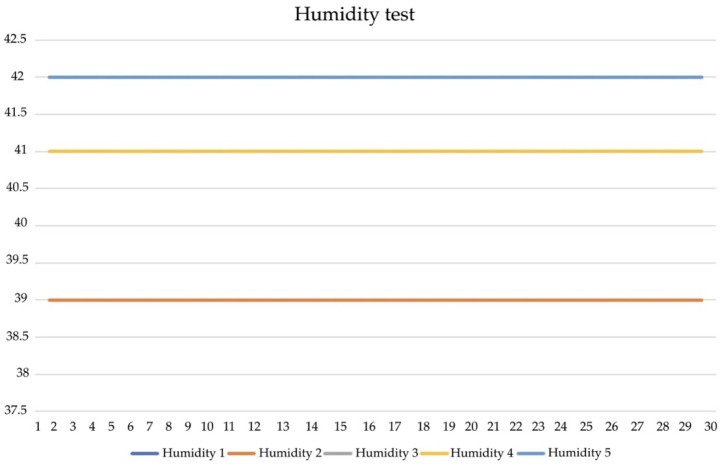
Humidity test.

**Figure 16 diagnostics-13-00145-f016:**
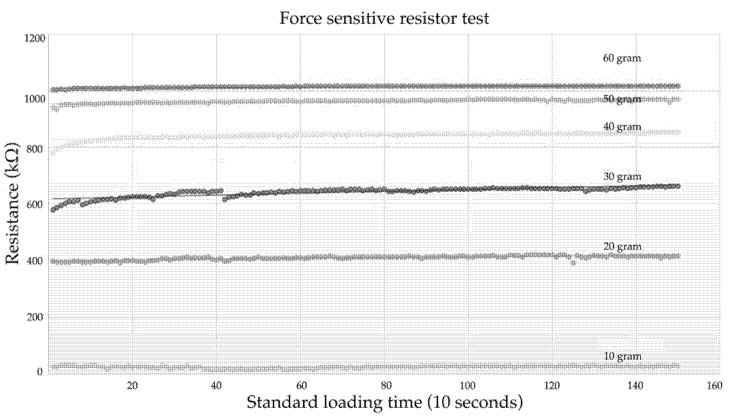
Resistance and standard loading time.

**Figure 17 diagnostics-13-00145-f017:**
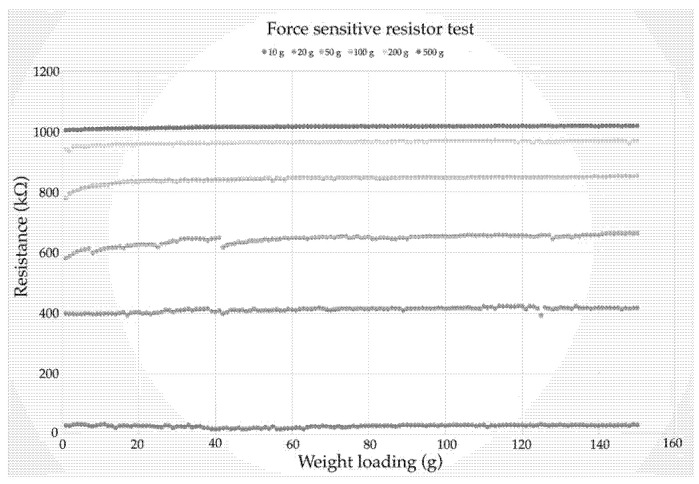
Resistance and weight loading test.

**Figure 18 diagnostics-13-00145-f018:**
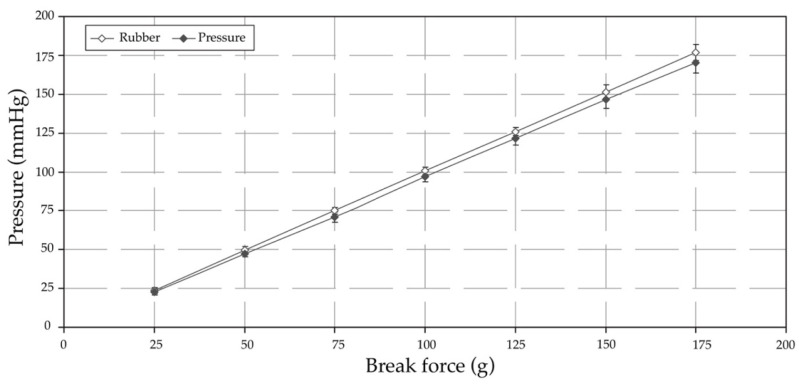
Static pressure sensor.

**Figure 19 diagnostics-13-00145-f019:**
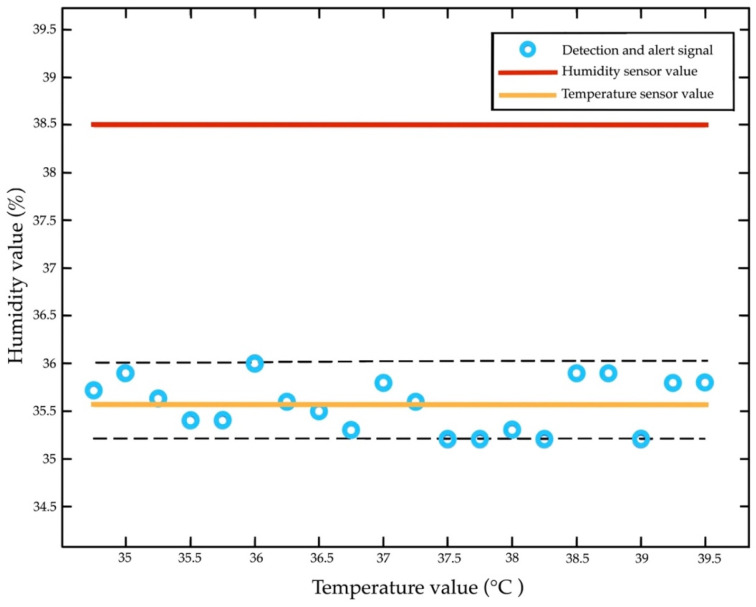
Repeatability of temperature and humidity values.

**Figure 20 diagnostics-13-00145-f020:**
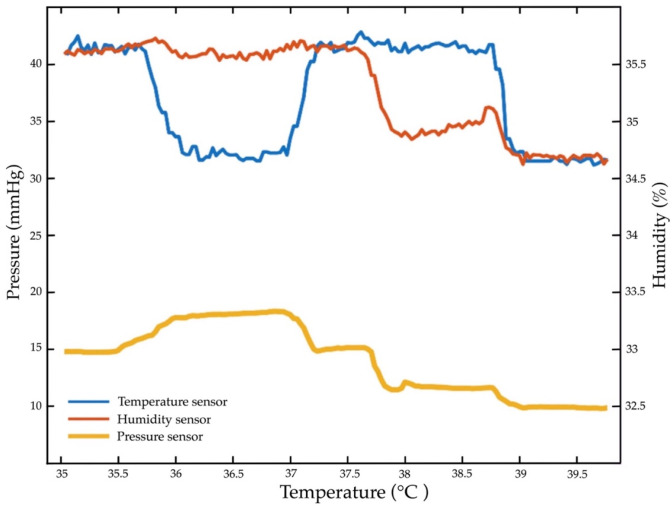
The repeatability test of the pressure, temperature, and humidity sensors.

**Table 1 diagnostics-13-00145-t001:** Experimental group.

No.	Internal Factors			External Factors			PI Risk	PI Incidence	PI Alert
	Gender	Age	Disease	Pressure (mmHg)	Humidity (%)	°C			
1	Male	60	DM + HT	32	30	37.2	High	No	Alert
2	Female	58	DM + HT+DLP	30	20	36.7	No	No	Normal
3	Male	75	DM + HT+DLP+AF	30	40	36.5	No	No	Beware
4	Female	88	DM + HT+DLP	30	50	36.7	Moderate	No	Alert
5	Female	90	DM + HT+DLP	34	30	37.1	High	No	Alert
6	Male	60	DM + HT	32	30	36.8	High	No	Alert
7	Female	58	DM + HT	25	20	36.7	No	No	Normal
8	Male	75	DM + HT + DLP + AF	30	20	37.2	No	No	Beware
9	Female	76	DM + HT + DLP	32	40	36.6	High	No	Alert
10	Female	90	DM + HT + DLP	34	30	36.5	High	No	Alert
11	Female	56	DM + HT	30	20	36.6	No	No	Beware
12	Male	48	DM + HT + DLP	34	30	36.5	Moderate	No	Alert
13	Male	66	DM + HT + DLP	34	40	36.5	High	No	Alert
14	Female	59	DM + HT	35	40	37.1	High	No	Alert
15	Female	66	DM + HT + DLP	35	20	37.2	High	No	Alert

Abbreviations: AF = atrial fibrillation; DM = diabetes mellitus; HT = hypertension.

**Table 2 diagnostics-13-00145-t002:** Control group.

No.	Internal Factors			External Factors			PI Risk	PI Incidence
	Gender	Age	Disease	Pressure (mmHg)	Humidity (%)	°C		
1	Male	63	DM + HT + DLP + AF	35	40	37.1	High	Yes
2	Female	58	DM + HT	25	30	36.3	No	No
3	Male	48	DM + HT + DLP	31	35	37.1	Low	No
4	Male	66	DM + HT + DLP	34	40	36.9	High	Yes
5	Female	59	DM + HT	34	30	37	High	Yes
6	Male	66	DM + HT + DLP	35	40	36.6	High	Yes
7	Female	54	DM + HT	30	20	36.8	No	No
8	Male	65	DM + HT + DLP + AF	31	20	37.2	Moderate	Yes
9	Female	70	DM + HT + DLP	30	50	36.6	Moderate	Yes
10	Female	82	DM + HT + DLP	34	30	37.1	High	Yes
11	Female	53	DM + HT	30	20	36.6	No	No
12	Male	50	DM + HT + DLP	30	20	36.7	No	Yes
13	Male	63	DM + HT + DLP	34	20	36.9	High	Yes
14	Female	61	DM + HT	34	40	37.2	High	Yes
15	Female	64	DM + HT + DLP	32	40	37.3	High	Yes

Abbreviations: AF = atrial fibrillation; DM = diabetes mellitus; HT = hypertension.

**Table 3 diagnostics-13-00145-t003:** The force-sensitive sensor of body detection.

	Body Area	Detected			
		Supine	Left-Lying	Right-Lying	Recall
Original	Supine	35	1	4	0.748
	Left-lying	9	27	0	0.719
	Right-lying	12	1	23	0.552
Precision	–	0.760	0.954	0.725	–

**Table 4 diagnostics-13-00145-t004:** Comparison of temperature and humidity data.

Temperature			Humidity		
Normal T (°C)	TT (Ω)	RTD (Ω)	Normal (%)	RH (%)	Sensor (Ω)
10	1038.8 ± 0.1	240.5 ± 0.0	10	7.5 ± 0.5	12,514.9 ± 11.2
15	1057.4 ± 0.1	244.2 ± 0.1	20	17.4 ± 0.6	12,689.8 ± 1.6
20	1076.2 ± 0.1	248.1 ± 0.1	30	27.2 ± 0.2	12,827.2 ± 2.6
25	1096.3 ± 0.2	252.3 ± 0.2	40	36.5 ± 0.3	12,935.5 ± 1.9
30	1114.9 ± 0.6	256.0 ± 0.2	50	45.8 ± 0.2	13,032.5 ± 2.7
35	1134.5 ± 1.3	260.1 ± 0.2	60	55.0 ± 0.4	13,140.9 ± 3.6
40	1153.7 ± 0.0	264.0 ± 0.0	70	64.4 ± 0.2	13,271.1 ± 5.1

**Table 5 diagnostics-13-00145-t005:** A comparison of averages, ranges, and loading times.

No.	SL	HL	SH	HH	$AV_{s}$	$RP_{s}$
1	930	450	660	270	560	2.57
2	770	530	520	360	680	2.07
3	540	470	550	330	510	1.70
4	720	540	600	360	510	2.58
5	700	390	570	350	500	1.88
6	850	530	670	540	400	2.00
7	450	360	490	450	520	1.65
8	700	500	530	370	580	2.19
9	710	550	360	410	460	2.16
10	760	610	420	540	380	1.97
${\mathrm{AV}}_{m}$	850	640	620	380	590	2.07
${\mathrm{RP}}_{m}$	1.69	1.47	1.25	1.31	1.48	–

**Table 6 diagnostics-13-00145-t006:** Conductive force-sensitive resistor.

Feature	Value
Nominal thicker	0.30 mm
Active sensor area	35.1 mm × 35.1 mm
	Semi-conductive layer: 0.10 mm/U1tem
Rubber mattress build	Spacer adhesive: 0.10 mm/Acrylic
	Conductive layer: 0.10 mm/U1tem
	Rear adhesive: 0.5 mm/Acrylic
Wide-force sensitive range	<100 g–1 kg
Break force (turn-on force)	20 g to 100 g
Stand-off resistance	200–1200 kΩ
Temperature operating range	35 °C to + 40 °C
Number of actuations (lifetime)	>10 million actuations

## Data Availability

Not applicable.

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
