# Peer review of "Electronic Alert Signal for Early Detection of Tissue Injuries in Patients: An Innovative Pressure Sensor Mattress"

_diagnostics, 2023, doi:10.3390/diagnostics13010145_

Round 1
Reviewer 1 Report
The writing problems in the article make it very difficult to understand. The abstract presents confusing sentences :"The electronic alert signal for early detection of tissue injuries is essential prevention of lightly pigmented skin, but there is little evidence for its effectiveness." If there is no evidence of its effectiveness, how can it be essential?
What the authors mean by this sentence? "The repeatability of sensing data indicated that 32 mmHg pressure, temperature (36°C), relative humidity (33.5%), response time (3 ms), loading time (30 g), and density areas (1mA)"
pg. 3 - 98 : A growing literature has defined four stages of tissue injuries: 10% (stage I), 15% (stage II), full-thickness skin to stage II and IV, may be inappropriate clinical care [37]. What about stage III?
pg 3 - 108 - [40] classified that temperature is cold and superficial heat for sensing data to sustain cutaneous detection of tissue injuries. confusing sentence
pg 3 - 118 what the authors mean by "mc"?
pg3. 127 - "Humidity sensing data is defined as friction of skin's surface" - Again, I am sure the authors doesn't mean this is the definition of humidity.
section 3 - Early detection of lightly pigmented skin damage is mechanically-deformed in the real time on the skin-integrated electronic sensors [53]. confusing sentence
"Figure 5 presents the functional sensing detection. " The authors should explain the details of figure in the text.
"Figure 7 presents the sensing control system. Force sensor sys-
tem is displayed in Figure 8. Sensing signal mechanism is depicted in Figure 9." The authors should explain the details of figure in the text.
Table 1 to 2: DM, HT...? please define
4.2 : "...precision ranging from 0 to 15 MPa" - ???? variable precision???
Fig 11 and 12 needs further text explanation
Table 3 - need figures to illustrate positions.
Fig 13 - Authors must interpret the results and write in the text.
4.4 - The reference of temperature sensor data was repeated ±0.5–2%??? What do the authors mean by this sentence?
authors should describe the results reported in table 5, figures 14 and 15.
4.7. Repeated Temperature, Humanity and Pressure SenSor Test
4.7 - The experimental test is one week to obtain the pressure loading ... confusing sentence
Fig. 19 is not called in the text.
Anyway, the writing really need improvements and actually avoid a better understanding of the article. Furthermore, the objectives of the article are not described. Nor are contributions reported. What's new in the article? What did the presented system improve in relation to others (similar) systems - what are the references that leaded the authors to the development of the system?
Author Response
Response to reviewer #1
Point 1:
The writing problems in the article make it very difficult to understand. The abstract presents confusing sentences: "The electronic alert signal for early detection of tissue injuries is essential prevention of lightly pigmented skin, but there is little evidence for its effectiveness." If there is no evidence of its effectiveness, how can it be essential?
Response 1:
Thank you for taking the time to carefully review our manuscript. We have carefully revised the manuscript which puts forward of your suggestion. In our revised manuscript, we have now clarified the abstract section, especially we have rewritten new version as “However, the electronic alert signal for early detection is limited due to the lack of pressure sensors to go along with visual assessment, particularly in patients with lightly pigmented skin injuries” (see line 16-18, page 1).
Point 2:
What the authors mean by this sentence? "The repeatability of sensing data indicated that 32 mmHg pressure, temperature (36°C), relative humidity (33.5%), response time (3 ms), loading time (30 g), and density areas (1mA)"
Response 2:
Thank you for your detailed revision of our manuscript. In our revised version, we have now rewritten and clarified as your suggestion. The new revised version as “The early detection of the pressure sensor to electronic alert signal at 32 mm Hg, a temperature of 37°C, a relative humidity of 33.5%, a response time of 10 seconds, a loading time of 30 g, density areas of 1mA, and resistance of 7.05 MPa (54 N) at 0.87/min” (see line 21-24, page 1).
Point 3:
- 3 - 98: A growing literature has defined four stages of tissue injuries: 10% (stage I), 15% (stage II), full-thickness skin to stage II and IV, may be inappropriate clinical care [37]. What about stage III?
Response 3:
We appreciated this insightful remark. In our revised article, we have intensively added and modified the sentence as “Indeed, as alluded to above, previous studies have defined four stages of tissue injuries: non-blanchable erythema of intact skin (stage I), partial-thickness skin loss with exposed dermis (stage II), full-thickness skin loss (stage III), and full-thickness tissue loss (stage IV)” (see line 91-94, page 3).
Point 4:
pg 3 - 108 - [40] classified that temperature is cold and superficial heat for sensing data to sustain cutaneous detection of tissue injuries. confusing sentence
Response 4:
Thank you for your detailed revision. In our revised version, we have modified as “According to Kokate et al.’s [40] definition, ambient temperature is detected on the skin surface of patients’ body organs, such as the heart, liver, brain, and blood” (see line 104-106, page 3).
Point 5:
pg 3 - 118 what the authors mean by "mc"?
Response 5:
Thank you for your comments. In our revised version, we have added the full term as “> 2 moisture” (see line 114, page 3).
Point 6:
pg3. 127 - "Humidity sensing data is defined as friction of skin's surface" - Again, I am sure the authors don’t mean this is the definition of humidity.
Response 6:
Thank you for this used observation. In our revised version, we have changed and modified as “Humidity in patients with tissue injuries is defined as subepidermal moisture-induced tissue damage of the water in the epidermal and dermal tissues [47]” (see line 122-123, page 3).
Point 7:
section 3 - Early detection of lightly pigmented skin damage is mechanically-deformed in the real time on the skin-integrated electronic sensors [53]. confusing sentence
Response 7:
Thank you for your remark. We have now modified as “The electronic signal alert procedures used to detect lightly pigmented skin injuries are combined with the pressure detection (mm Hg), temperature (T), and humidity (H) [53]” (See line 139-140, page 3)
Point 8:
"Figure 5 presents the functional sensing detection. " The authors should explain the details of figure in the text.
Response 8:
Thank you for your suggestion. We have rewritten as “Figure 5 details the functional detection of the pressure sensor mattress. The mattress array included four top and four bottom force sensors, all of which measure the patient's weight. All sensors received the detection data, which was then displayed on the microcontroller. The sensor database was subsequently sent to the dashboard monitor of the pressure sensor mattress” (see line 175-179, page 5).
Point 9:
"Figure 7 presents the sensing control system. Force sensor system is displayed in Figure 8. Sensing signal mechanism is depicted in Figure 9." The authors should explain the details of figure in the text.
Response 9:
Thank you for highlighting this. In our revised version, we have modified as “Figure 7 depicts the control system of the signal mechanism. The system consists of two signal sensors: four temperature and humidity sensors and eight pressure sensors, all of which are connected to the microcontroller monitor. The temperature and humidity sensor and signalling data are sent to the electronic alert system. The force-sensitive system is displayed in Figure 8. The sensor is mounted on the FSR sensor, which measures the interface pressure of the patient’s side-lying position. The FSR data is determined by the change of the electrical voltage. Data is sent to the pressure sensor mattress via the dashboard monitor reporting.” (see line 196-213, page 6).
Point 10:
Table 1 to 2: DM, HT...? please define
Response 10:
Thank you for your observation. We have written as “AF = atrial fibrillation; DM = diabetes mellitus; HT = hypertension” (see the abbreviation of Table 1 and 2).
Point 11:
4.2 : "...precision ranging from 0 to 15 MPa" - ???? variable precision???
Response 11:
Thank you for highlighting this. In our revised version, we have modified as “The experimental test ranged from 1–15 MPa for the online converter of weights, which was used to measure patients with tissue injuries” (line 251-252, page 8).
Point 12:
Fig 11 and 12 needs further text explanation
Response 12:
Thank you for your detailed revision of our manuscript. In our revised version, we have added the further text explanation as “Figure 10 depicts the platform of the pressure sensor mattress. The platform was tested using patient information (gender, age, ID code, and disease) records on the dashboard monitor. The system of pressure detection, temperature and humidity sensors, which collected to set a timer change and occur to the patient information. Figure 11 presents the structure of the dashboard monitor. The screen displays both pressure detection and temperature and humidity sensors, which are linked to the electronic alert signal on the dashboard monitor. The platform monitor (turn-on and off) can take the form of an automatic system, with a timer, turn left and right, and/or a handling monitor screen” (see line 257-264, page 9).
Point 13:
Table 3 - need figures to illustrate positions.
Response 13:
Thank you for your suggestion. We have modified the table 3 into figure 12 and then future text explanation of “Figure 12 illustrates the body area sensors of the experimental group. The positional sensor is the supine area of the body’s angle (45° from the body, midstance 15° femur, sagittal angle 30° dorsal, and frontal angle 45° medial). The body sensors of the left-lying and right-lying positions are set at 90° from the body sensor (sagittal angle 15° ventral, frontal angle 30° medial, and sagittal angle 45° dorsal). The force-sensitive detection is presented in Table 3. (see Figure 12 and line 279-281, page 9).
Point 14:
Fig 13 - Authors must interpret the results and write in the text.
Response 14:
Thank you for your detailed revision. We have interpreted the results of Figure 13 as “The experimental test using weights can be seen in Figure 13. Regarding the kΩ, it was shown that the fast response time appeared higher in sensors 2 and 3, ranging from 2.59–5.83% of probability. There was only a 1.44% probability that sensors 1, 4, and 5 would be misclassified as sensor 6. FSR drifts were plotted on the loading pressure from 5–10 g, and were applied to the sensor for 10 seconds. There are almost 5 times when compared to pressure loading (30 g), allowing the system to monitor respiratory rates up to 1,000 k per 10 seconds” (line 292-298, page 10).
Point 15:
4.4 - The reference of temperature sensor data was repeated ±0.5–2%??? What do the authors mean by this sentence?
Response 15:
We appreciated this insightful remark. In our revised version, we have intensively modified as suggested the terms of “The reference of the temperature data was repeated from 0.5–2% of the relative humidity (see line 305, page 11).
Point 16:
authors should describe the results reported in table 5, figures 14 and 15.
Response 16:
We really appreciate this kindly reminding. We have now explained the text of table 5 as “Table 4 compares the temperature and humidity sensors. The normal temperature remains stable at 40°C and was recorded every 10 seconds; this was repeated for the RTD240.5 ± 0.0 reference sensor. The response time of RTD was more than 10 times that of 264.0 ± 0.0, with the average 1/ (30 seconds) during the mechanical ventilation. The values obtained showed that the H steps of 10% RH were then generated to 10–70% RH, keeping T constant at 7.5 ± 0.5°C. We also compared the variability of the reference to RTD260.1 ± 0.2 of T (35°C) at RH55.0 ± 0.4, defined as the time to get from 10% to 70% of the final values. (see line 308-314, page 11).
Moreover, we have explained in figures 14 and 15 reporting as “Figure 14 illustrates the temperature test. The temperature was recorded using a high-resistance probe sensor placed 10 cm into the rectum, with an accuracy of +0.2°C, and 3-point calibration. Based on this measurement, sensors 1 and 2 of T1, T2, and T3 were placed in areas of the body (lying position); the sensor achieved detection of 35.5°C, and 27 mm thickness. The T4 and T5 of sensors 3 and 4 on the disk are usually detected accurately (within ±10%) and tolerate a temperature gradient up to 37.5°C. The results (Figure 15) highlight that the humidity remained constant at 36.5% and moduli ranged from 37.5–42.5% RH. The validity of the humidity outputs was monitored using a hygrometer (±5% accuracy) of 64.4 ± 0.2, which increases by 42.5% every 30 seconds, respectively” (see line 317-325, page 12).
Point 17:
4.7. Repeated Temperature, Humanity and Pressure Sensor Test
4.7 - The experimental test is one week to obtain the pressure loading ... confusing sentence
Response 17:
Thank you for your remark. In the revised manuscript has been revised as “The sensor detection is the core pressure loading of 32 mm Hg, T (30–40°C), and RH (30–40%)” (see line 403-404, page 15).
Point 18:
Fig. 19 is not called in the text.
Response 18:
Thank you for your observation. We have added the number figure 19 in the text (see line 408, page 15).
Point 19:
Anyway, the writing really needs improvements and actually avoid a better understanding of the article. Furthermore, the objectives of the article are not described. Nor are contributions reported. What's new in the article? What did the presented system improve in relation to others (similar) systems - what are the references that leaded the authors to the development of the system?
Response 19:
Thank you for this comment and for pointing out that we have to clarify this further. We have extensively modified the manuscript in order to fulfil your expectations.
First, we have extensively edited as a whole of the manuscript to make easy understanding of writing and interpretation.
Second, the quality of communication has been entirely improved by the Cambridge Proofreading LLC. The manuscript was edited for proper UK English, grammar, punctuation, spelling, and overall style by two academic editors, namely, Dr. Michael M and Dr. Powell, who are a native language editing.
Third, we have now described the objective of the study as “The purpose of the current study is to develop an innovative pressure sensor mattress using an electronic signal alert for early detection of tissue injuries. We aim to examine perfusion and blood circulation (wet and sweat) using the temperature (°C) and humanity (%) sensors” (see line 69-72, page 2).
Finally, we have extensively modified and clarified as a whole of the manuscript (see our revised version using track changes by each sentence-to-sentence).

Reviewer 2 Report
This manuscript collects patient-related data through multiple electronic sensors in the mattress for early warning of skin damage. The manuscript uses pressure sensors combined with temperature and humidity sensors, and a wireless data transmission and analysis system is built. Relevant data ranges and thresholds are discussed by analyzing the accuracy and error of the collection system. The system has potential applications in patient care and health monitoring. Few comments are outlined below.
1.Although the patient experimental data are relatively small, they are worthy of further analysis and discussion.
2.As Figure 15 and Figure 17, the results are not shown clearly enough.
3.The authors have done a lot of work, but not enough to summarize the innovation. I hope the innovative point of the work can be further highlighted.
Author Response
Response to reviewer #2
Point 1:
This manuscript collects patient-related data through multiple electronic sensors in the mattress for early warning of skin damage. The manuscript uses pressure sensors combined with temperature and humidity sensors, and a wireless data transmission and analysis system is built. Relevant data ranges and thresholds are discussed by analysing the accuracy and error of the collection system. The system has potential applications in patient care and health monitoring. Few comments are outlined below.
Response 1:
Thank you for taking the time to carefully review our manuscript. We have extensively modified the manuscript in order to fulfil your expectations. We believe the manuscript is now stronger and clearer.
Point 2:
Although the patient experimental data are relatively small, they are worthy of further analysis and discussion.
Response 2:
Thank you for your detailed revision of our manuscript. We have extensively modified the patient experimental data of further discussion and analysis (see our discussion section, page 15-16).
Point 3:
As Figure 15 and Figure 17, the results are not shown clearly enough.
Response 3:
Thank you for your suggested revision of our manuscript. In our revised manuscript, we have extensively added the further figure 15 results as “The results (Figure 15) highlight that the humidity remained constant at 36.5% and moduli ranged from 37.5–42.5% RH. The validity of the humidity outputs was monitored using a hygrometer (±5% accuracy) of 64.4 ± 0.2, which increases by 42.5% every 30 seconds, respectively” (see line 314-317, page 11).
Point 4:
The authors have done a lot of work, but not enough to summarize the innovation. I hope the innovative point of the work can be further highlighted.
Response 4:
Thank you very much for this constructive suggestion and detailed revision of our manuscript. In our revised version, we have now clarified the innovation as a whole of the work, especially our conclusion section “The purpose of the current study is to develop an innovative pressure sensor mattress using an electronic signal alert for early detection of tissue injuries. We aim to examine perfusion and blood circulation (wet and sweat) using the temperature (°C) and humanity (%) sensors” (see line 457-462, page 16).
Round 2
Reviewer 1 Report
I have found some small English problems. In the opinion of this review the paper can be accepted after minor corrections.
Author Response
Point 1: I have found some small English problems. In the opinion of this review the paper can be accepted after minor corrections.
Response 1: Thank you for taking the time to carefully review our manuscript. Our manuscript has been entirely improved by two academic editors, namely Dr. Daniel Mager and Dr. Danny Jackson who is a professional language editing.
Reviewer 2 Report
In the revised version, the author added the missing information and revised the content, At the same time, the more result discussion was supplemented. Apart from a lack of innovation, the work of the manuscript deserves to be published.
